# Blue Vane and Pan Traps Are More Effective for Profiling Multiple Facets of Bee Diversity in Subtropical Forests

**DOI:** 10.3390/insects15110909

**Published:** 2024-11-20

**Authors:** Ting-Ting Xie, Ming-Qiang Wang, Yi Li, Cheng-Yong Su, Dan Zhang, Qing-Song Zhou, Ze-Qing Niu, Feng Yuan, Xiu-Wei Liu, Ke-Ping Ma, Chao-Dong Zhu, Jia-Sheng Hao, Douglas Chesters

**Affiliations:** 1College of Life Sciences, Anhui Normal University, 1 Beijing East Road, Jinghu District, Wuhu 241000, China; tingtingxie2021@ioz.ac.cn (T.-T.X.); sky475342@163.com (C.-Y.S.); 2Key Laboratory of Zoological Systematics and Evolution, Institute of Zoology, Chinese Academy of Sciences, 1 Beichen West Road, Chaoyang District, Beijing 100101, China; zhouqingsong@ioz.ac.cn (Q.-S.Z.); niuzq@ioz.ac.cn (Z.-Q.N.); yuanf@ioz.ac.cn (F.Y.); zhucd@ioz.ac.cn (C.-D.Z.); 3Key Laboratory of Mountain Ecological Restoration and Bioresource Utilization of Sichuan Province, Chengdu Institute of Biology, Chinese Academy of Sciences, 4 Renmin South Road, Wuhou District, Chengdu 610041, China; wangmq@cib.ac.cn; 4Key Laboratory of Ecological Restoration Biodiversity Conservation of Sichuan Province, Chengdu Institute of Biology, Chinese Academy of Sciences, 4 Renmin South Road, Wuhou District, Chengdu 610041, China; 5State Key Laboratory of Vegetation and Environmental Change, Institute of Botany, Chinese Academy of Sciences, Beijing 100093, China; liyi2021@ibcas.ac.cn (Y.L.); kpma@ibcas.ac.cn (K.-P.M.); 6Characteristic Laboratory of Forensic Science in Universities of Shandong Province, Shandong University of Political Science and Law, Jinan 250014, China; zhangdanioz2016@gmail.com; 7Institute of Agro-Products Processing, Xueyun Road, Kunming 650221, China; liuxiuwei0305@hotmail.com; 8College of Life Sciences, University of Chinese Academy of Sciences, Beijing 100049, China; 9Zhejiang Qianjiangyuan Forest Biodiversity National Observation and Research Station, Institute of Botany, Chinese Academy of Sciences, Beijing 100093, China; 10China National Botanical Garden, Beijing 100093, China; 11International College, University of Chinese Academy of Sciences, Yuquan Road, Shijingshan District, Beijing 100049, China

**Keywords:** bee diversity, biodiversity monitoring, blue vane traps, DNA barcoding, forest insects, pan traps, pollinator monitoring

## Abstract

Little attention has been paid to the efficiency of trap types in capturing bees across taxonomic and functional groups, nor their suitability under varying environmental conditions. Our study evaluated the efficiency, bias, and complementarity of four trap types: yellow, white, and blue pan traps, and blue vane traps for pollinator monitoring in monoculture and mixed forests. We found that bias in trap types was not only detected in taxonomic but also in functional groups. Differences in bee taxonomic, phylogenetic, and functional diversity were also observed, with blue pan traps yielding the highest species richness and phylogenetic diversity, while blue vane traps captured the highest functional richness. When considering the complementarity of different traps, the combination of blue pan and vane traps outperformed the other two-method combinations. Notably, the bias in trap types was most pronounced in mixed forests.

## 1. Introduction

Bees, particularly wild bees, are one of the most important pollinators of plants [1,2,3], contributing more to pollination efficiency than other flower-visiting animals [1,4,5]. However, bees are highly sensitive to environmental changes [6,7,8] and have experienced significant declines in the past decades due to habitat loss and fragmentation, land use intensification, climate change, pathogens, and pesticides [9,10,11,12]. Understanding how bees respond to various stressors is essential for developing effective conservation policies and for management, to ensure the long-term stability of ecosystems [13,14,15]. To achieve accurate and reliable results and reduce the incidence of bias, it is crucial to employ sampling methods that are accurate, practical, and repeatable [16,17,18]. Accurate assessments of bee diversity are an important foundation for effective conservation and management.

Bee sampling methods can be classified as active and passive [17,18]. Active methods are mostly ‘seek and collect’ by observers, and take place in a given area and period [19]. In contrast, passive traps are left in a target area for a given period and accumulate captured insects over time (e.g., Malaise, pan and vane traps) [20]. There are strengths and weaknesses of passive and active collection methods [17,18]. The efficiency of activate methods, particularly sweep netting, depends on the experience and professionalism of collectors [21]. For instance, O’Connor et al. (2019) found that sweep-netters with greater taxonomic expertise could produce equivalent species accumulation data as those obtained from pan traps [21]. The efficiency of collection by sweep netting is also affected by topography, which is frequently heterogeneous and complex in forests [22]. Passive methods, including vane traps, colored pan traps, and Malaise traps, have been widely used for collecting flying insects in various habitats, including wild bees [14,23,24]. Vane and pan traps attract bees by imitating the color and shape of flowers, and they are relatively inexpensive and labor-efficient compared to active methods. Malaise traps are flight interceptors and more suited to long-term monitoring, though they come at a considerable price, their establishment is laborious and the proportion of bees amongst flying insects captured is lower than other trap options [25,26]. Passive traps can be more versatile, as they can be applied in various environments and avoid collector bias. However, passive traps nonetheless have a preference for certain taxa [27]. For example, stronger-flying bees are known to escape from pan traps [28], while vane traps capture a higher proportion of larger-bodied bees [29].

Comparisons of insect trapping methods have primarily focused on single measurements of diversity, usually species richness and composition [30,31,32,33]. However, this may ignore the preferential attraction of the sampling method to different taxonomic and functional groups [27,34,35]. Such biases can be significant, given that traits, including body size, diet breadth, and proboscis length, play important roles in biodiversity maintenance and ecosystem functions [36,37], and mediate various behaviors. Any sampling bias in these types of traits might obscure environmental effects on bee community composition. Furthermore, it remains unclear whether capture bias is reflected in commonly used diversity indices. Most comparisons of trap types consider only species richness [38,39], frequently also used to elucidate community diversity in response to environmental changes [40,41]. However, phylogenetic diversity has gained importance in biodiversity conservation [42] because it captures the uniqueness of lineages and reveals various ecological pressures via a more nuanced description of community structure [43,44,45]. Moreover, functional diversity has been shown to be more sensitive to environmental changes than species richness. For example, the decrease in functionality was greater than what would be expected based solely on the reduction in species richness due to the preferential loss of functionally distinct species [46]. Therefore, incorporating these three dimensions of diversity—taxonomic diversity, phylogenetic diversity, and functional diversity—can provide a more comprehensive understanding of observed patterns in diversity.

The methodology used in comparing sampling methods has often overlooked scale and context [33,47], as it has been shown that traps do not perform consistently across habitats [48,49]. For example, a comparison of several trap types found efficiency varied across habitats, with blue pan traps being more effective in natural sites than in orchards, while an opposing trend was observed for yellow pan traps [50]. Similarly, bees caught in pan traps have been found to decrease with increasing competition in floral resources [30], whereas blue vane traps proved effective even with intense competition for floral resources [51]. However, only limited attention has been given to sampling methods for bees in forests, a critically important hotspot of biodiversity [52]. Furthermore, no comparative analyses have been carried out across forest diversity gradients, a variable that accounts for considerable variance in primary productivity and maintenance of forest-associated biodiversity [53,54,55].

Here, we addressed these gaps by comparing the attractiveness of various trap types to bees with respect to taxonomic and functional groups and across different forest diversity levels. Additionally, we compared three dimensions of diversity (taxonomic, phylogenetic, and functional diversity) yielded by different trap types. We hypothesized that (i) different trap types differed in their bias in specific taxonomic and functional groups, (ii) such bias is revealed by diversity variation, and (iii) the sampling bias across trap types changes in different forest habitats.

## 2. Materials and Methods

### 2.1. Study Area, Bee Collection, and Processing

The study was conducted at the Biodiversity-Ecosystem Functioning (BEF) experiment located in Jiangxi Province, southeast China (29°080–29°110 N, 117°900–117°930 E, Appendix A) [56]. This experiment comprises two sites that were established in 2009 and 2010. The climate is subtropical with a mean annual temperature of 16.7 °C and mean annual precipitation of 1821 mm [57]. Our study included a total of 58 plots from sites A and B, including 28 monoculture plots with one tree species per plot and 30 plots with two or more tree species per plot. 

The traps (colored blue, yellow, and white) were selected for bee collection in our study because they have been proven to be efficient in collecting a wide variety of bees in other ecosystems [27,38,58]. In each plot, we established two sets of blue vane traps and three-colored (yellow, blue, and white) pan traps, arranged diagonally (Appendix A). This configuration resulted in a total of eight traps per plot. Across all 58 plots, we placed 464 traps for each sampling day. To minimize biases resulting from trap placement within each plot, we adopted a systematic approach rather than random selection. The placement of each trap inside the plot was in a fixed arrangement with blue vane traps placed in the northwest and southeast direction, and pan traps placed in the northeast and southwest direction. Additionally, the plots were randomly located in the forests, which helped to ensure that any bias associated with fixed trap placement within a plot would be mitigated by the variable environmental conditions and microhabitats encountered across forest plots. The traps were uniformly positioned in the understory with blue vane traps suspended at a height of 1 m, and the pan traps positioned at a height of 0.5 m (Appendix A), taking into consideration the dense vegetation in the understory of some plots. Bees were collected during three periods widely known for bee activity and flowering: June (summer) and September (early autumn) of 2022, and April (spring) of 2023 to minimize the differences in bee activity that might be caused by seasonal variation. Sampling was conducted every 24 h, three times per sampling event, resulting in 9 sampling days in total. All the samples were stored in 99.9% ethanol in the field. In the lab, bees were sorted and then the specimens were pinned. The pinned bees were subsequently morphologically examined. At a minimum, 5 specimens from each morphospecies were randomly selected, and their mid-right legs were carefully extracted for molecular work [59].

COI barcodes were obtained following the pipeline described by Liu et al. (2017), including four main steps: DNA extraction, PCR amplification, molecular delimitation, and taxonomic assignments [59]. The DNA was extracted by TIANGEN Guide Smart DNA extraction kits (TIANGEN BIOTECH Beijing Co., Ltd., Beijing, China) and sequencing was conducted at Beijing Tianyi Huiyuan Biotechnology Co., Ltd. (Beijing, China). Haplotypes were inferred using Mothur v1.40.3 [60] and then molecular species level delimitation conducted with five tools: Mothur v1.40.3 [60], CD-Hit v4.8.1 [61], bPTP v0.51 [62], the Vsearch v2.13.3 ‘cluster_fast’ function [63], and Blastclust v2.2.12. Molecular Operational Taxonomic Units (MOTUs) were assigned taxonomic names with the software SAP v1.9.9 [64], using reference DNA barcodes downloaded from the BOLD system at https://www.boldsystems.org/ (accessed on 8 April 2022) [65]; Taxonomic assignment was conducted using the command ‘--assignment ConstrainedNJ’ with the minimum identity set as 0.92 (‘--minidentity 0.92’). Specimen taxonomic identities were also confirmed via morphological inspection with the help of taxonomists, ensuring identification at least to the genus level. Species names were finalized considering both the molecular assignment and morphological identification. Frequently, a single species exhibited multiple COI sequences due to genetic variation. To streamline our analytical approach, we used the most prevalent sequences for each species to proceed with subsequent analyses. This selection included the construction of a phylogeny and the calculation of phylogenetic diversity indices, ensuring that our findings were based on the most robust and representative genetic data.

### 2.2. Bee Functional Traits

We selected three life-history traits (parasitism, sociality, and nesting location; for further details on trait categories, see Appendix A), and four morphological traits (inter-tegular distance, head width, forewing length, and body length), which are thought to be related to capabilities of obtaining pollen and sensitivity to environmental changes [14,66,67,68,69,70,71,72,73]. The life-history traits were obtained using the pipeline described in [74]. The pipeline was used to predict states for the set of 63 queries, using a phylogeny-based model of 2391 reference species and 3812 trait records. The reference phylogeny used in trait modeling was taken from the Insect Phylogeny synthesis hub at https://insectphylo.org/, accessed on 6 November 2023 [75]. The morphological traits were measured using a Zeiss Discovery V20 stereomicroscope (ZEISS AG, Oberkochen, Germany).

### 2.3. Statistical Analyses

To investigate the potential biases and visualize the distribution of bee species across trap types, taxonomic trees were plotted using the ‘metacoder’ package in R v4.3.3 [76]. Furthermore, to understand how sampling methods were biased with respect to functional traits, we compared the attractiveness to different functional traits. Specifically, we employed Chi-square tests to evaluate the associations between trap types and categorical traits (life-history traits: sociality, parasitism, and nesting location) and plotted results in R package ‘ggstatsplot’ with function ‘ggbarstats’ [77]. Additionally, we used one-tailed Wilcoxon tests with Benjamini–Hochberg (BH) adjustment [78] to compare the attractiveness of different trap types with continuous traits [79].

To test whether the bias was consistent with variation in diversity, we conducted a one-tailed pairwise Wilcoxon test with BH adjustment [79], comparing bee alpha diversity across the four trap types. Indices for three dimensions of diversity (species richness: taxonomic diversity, TD; Faith’s phylogenetic diversity: phylogenetic diversity, PD; and functional richness: functional diversity, FD) were calculated by using R package ‘vegan’ [80], ‘picante’ [81], and ‘FD’ [82], respectively. TD evaluates the count of unique species within the community [83], PD quantifies diversity by summing the length of branches between members on a phylogenetic tree [42], and FD measures the functional space (e.g., the range between the maximum and minimum value in the case of a single trait [84,85]) occupied by the community [86]. FD was evaluated based on four morphological and three life-history traits. The dimensionality of four newly measured traits was reduced through principal coordinate analysis (PCoA), using the first principal components to represent bee body size in the R function ‘pcoa’ from the R ‘ape’ package [87]. Due to the life-history traits used being categorical, a distance matrix that contained the functional distance for each pair of species was calculated according to ‘gower’ distance [88] in R package ‘FD’ with function ‘gowdis’ [82]. These diversity indices were calculated for each plot per month, treating different sampling events in different months as replicates. Due to inconsistency in success in attracting bees, not all traps within a given plot contained observations. As a result, the number of plots with the presence of bees differed across trap types. To maintain consistency in the level of sampling and ensure paired comparisons, the number of plots was kept equal to that of the more efficient trap types and we only omitted from analysis those plots where no bees were captured by any of the four traps.

To estimate completeness across sampling units, we performed a sample-size-based rarefaction and extrapolation sampling curve. The Hill diversity metric (q = 0) was computed to examine differences in three dimensions of diversity (TD, PD, and FD) in the R package ‘iNEXT.3D’ [89]. Given the preliminary analysis indicating limitations when using individual traps, we further investigated combinations of trap types to determine which are more likely to yield bee communities similar to those collected in a comprehensive survey. To evaluate dissimilarities between trap types and the complete survey, we employed the Mantel statistics for matrix correlations in the R package ‘vegan’ [80]. The Mantel test was performed by matrix rank correlations based on Spearman’s correlation coefficient with 999 permutations. In order to investigate whether different habitats would affect the sampling efficiency of trap types in the forests, all the analyses were conducted separately based on data obtained from monoculture or mixed plots (note, the mixed forests included plots with two or more tree species). In addition, we tested the overall sampling efficiency across all levels of tree diversity (named all forests hereafter).

## 3. Results

A total of 3993 bee specimens were collected and 1237 barcodes were obtained after sorting plot samples into morphospecies. The clustering tool CD-Hit v4.8.1 showed the most consistent results and thus the MOTUs resulting from this were used in further analyses. After morphological assessment, we found bees belonging to 5 families, 12 genera, and 63 species, with the most abundant family being Halictidae (Appendix A). In monoculture forests, 2230 specimens were obtained (4 families, 11 genera, and 57 species), while in mixed forests, 1763 specimens were collected (5 families, 12 genera, and 52 species).

The four trap types exhibited no significant differences in capturing bees from Colletidae, Andrenidae, and Megachilidae. However, blue pan traps exhibited a higher attraction for Halictidae bees compared to blue vane traps, while the latter showed a greater attractiveness to Apidae bees than three-colored pan traps (Figure 1a and Appendix A). The bee preferences of different trap types were not consistent across forest diversity levels. For example, in mixed forests, blue vane traps captured more Apidae bees than blue pan traps, while there was no significant difference in monoculture forests (Figure 1b,c and Appendix A).

The bees captured by blue vane traps were significantly larger in size compared to those trapped by three-colored pan traps (*p* < 0.01), with no significant difference in ITD observed among three-colored pan traps (*p* > 0.05, Figure 2). Approximately 6% were cleptoparasitic bees, 31% were above-ground nesters, and 12% were eusocial bees. The probability of collecting parasitic bees was found to be independent of trap types (Appendix A), while all traps were more effective in capturing bees that nested underground than aboveground (Appendix A). There was a correlation between trap type and sociality, but such a correlation was not observed in monoculture forests (Appendix A).

Blue pan traps yielded the highest species richness and phylogenetic diversity (*p* < 0.01), followed by blue vane traps (*p* < 0.05), while blue vane traps captured the highest functional richness (*p* < 0.01), with blue pan traps a close second (*p* < 0.01; Table 1). However, the efficiency varied in monoculture and mixed forests. In monoculture forests, the advantage of blue vane traps in terms of functional richness vanished compared with blue pan traps (*p* > 0.05; Appendix A; Appendix A). Similarly, the superiority of blue vane traps over yellow pan traps was not detected in the monoculture forests if considering species richness (*p* > 0.05; Appendix A; Appendix A). 

The sampling coverage completeness for all three facets of bee diversity was consistently greater than 0.95 for each trap type (Appendix A). Blue pan traps achieved the highest degree of completeness (sample coverage index = 0.98), while the white pan traps yielded the lowest (sample coverage index = 0.95). The sampling coverage index varied between monoculture and mixed forests.

Each trap showed specific unique biases towards taxonomic or functional groups, and also displayed distinct diversity profiles. This raised the question of whether the traps can complement each other in capturing a more comprehensive spectrum of bee diversity. To test this, we computed the Mantel statistical analysis, which indicated that different combinations of trap types could yield a concordance ranging from 0.84 to 0.99, relative to the complete survey. In general, combinations of three trap types performed better in matching the complete survey than combinations of two trap types, with one exception (three-colored pan traps, as shown in Appendix A). In terms of combinations of two trap types, the combination of the blue vane and pan traps demonstrated a very high concordance with the complete data set (r_M_ > 0.93, *p* < 0.05). Among three-method combinations, the combination of blue vane, blue pan, and yellow pan traps yielded nearly perfect concordance against the complete survey with a correlation coefficient exceeding 0.98 (Appendix A). The results of the correlation analyses differed for some combinations in different forests. For instance, the correlation calculated by yellow and white pan traps differed a lot in different forest diversity levels (monoculture forests: r_M_ = 0.88 and mixed forests: r_M_ = 0.78; *p* < 0.05).

## 4. Discussion

We used four passive traps to capture wild bees and compared their taxonomic and functional bias, as well as their efficiency in three facets of diversity across different forest types. Our results indicated bias both in terms of taxonomy and function across four trap types. Blue pan traps yielded the highest taxonomic (quantified by species richness) and phylogenetic diversity (assessed via Faith’s phylogenetic diversity index), while functional diversity (measured by functional richness) captured by blue vane traps was the highest. We also examined the complementarity among different trap types and discovered that the combinations of blue vane, blue pan, and yellow pan traps yielded the most comprehensive community against complete sampling. Notably, the performance of each trap depended on the forest type.

### 4.1. Effectiveness of Different Trap Types

In our assessments of bias and diversity capture, blue pan traps were found to be more efficient in evaluating bee diversity, with blue vane traps a close second in most cases. This is consistent with some previous comparisons. The color blue, which has a relatively short wavelength, has long been known to be discernable by various Hymenoptera [33,90,91,92]. However, blue pan traps captured more bees than blue vane traps, in contrast with the finding of previous findings [18,47]. In our study, three-colored pan traps were simultaneously set up and positioned a distance from blue vane traps, in each plot. Therefore, the overall attractiveness of three-colored pan traps might contribute to drawing more bees into the area [93], potentially enhancing the efficiency of the blue pan traps. In addition, there were variations in elevation between the pan and vane trap types, which could have inadvertently introduced bias into the composition of bee species captured, although our traps were consistently positioned at the suitable height for bee capture in the understory providing a consistent basis for comparison. Consequently, we suggest that future research take into account the height at which traps are set.

### 4.2. Effectiveness in Different Forest Diversity Levels

Our study revealed that the effectiveness of trap types varied across forest diversity levels, with the most pronounced differences observed in mixed forests. These differences were evident not only in the biases towards specific taxonomic and functional groups but also in the overall diversity captured. The rarefaction curve results also confirmed the differential sampling completeness achieved by the traps in different forests. Previous findings showed that tree diversity was positively associated with a diversity of understory herbs [40], indicating that mixed forests may accommodate a more complex understory structure and microclimate condition [94,95]. The complex structure in mixed forests might amplify the bias of trap types, thereby influencing effectiveness. However, the mechanism of how complex environmental conditions influence the sampling efficiency of trap types is unclear. We suggest that further research is needed to elucidate the underlying mechanisms governing the differences in trap bias and efficiency across various habitats, such as the level of tree species diversity.

### 4.3. Bias and Complementarity Among Trap Types

Previous studies have focused only on the comparisons between trap types, while ignoring bias and complementarity [21,26,27]. The rarefaction curve analysis indicated that our sampling efforts were sufficient and adequate to reflect total species diversity. Despite this, we observed significant biases in taxonomic and functional group composition among most trap types, and such bias could result in different community compositions. For instance, *Lasioglossum* species were predominantly captured in pan traps, as noted in previous studies [39,96]. Similarly, *Ceratina* species were attracted by blue vane traps, in agreement with the results of Campbell et al. (2023) [47]. These preferences indicate that trap types will inherently attract specific bee species, a phenomenon also observed in studies of other arthropods, particularly ants and spiders [32,49]. Our findings, consistent with those of O’Connor et al. (2019) [21] and Chamorro et al. (2022) [30], indicate that reliance on a single trapping method might lead to biases towards specific taxonomic and functional groups, thereby failing to provide a comprehensive representation of the bee community. Importantly, our findings also found that all trap types used were capable of capturing species that remained undetected by alternative traps, highlighting the value of employing multiple trap types to mitigate biases and gain a more accurate assessment of community composition. However, it should be noted that due to their high efficiency and complementarity, passive traps must be used carefully to prevent localized extirpation [97].

Given the incomplete capture of bee communities by any single trap type, we further investigated the efficiency of various combinations of trap types to explore complementarity effects. Surprisingly, the combinations of blue vane and pan traps consistently outperformed other two-method and one three-method combinations, in terms of community composition. The most effective two-method combination included the two most efficient trap types identified in our study. This means that these two traps often captured different bees from different phylogenetic clades and with different traits, and thus, would complement each other. We also examined the effectiveness of three-method combinations. The concordance to complete the survey was generally improved with the addition of trap types, except when the three-colored pan traps were used in combination.

Contrary to the widely held belief that three-colored pan traps are complementary for collecting and monitoring arthropod diversity, particularly bees [58], our study found that the most effective combination for capturing bee communities was the use of blue vane, blue pan, and yellow pan traps. This combination achieved the highest correlation with a complete survey. Giles and Ascher (2006) found that species richness of fast and highflying species such as *Megachile*, *Colletes*, and *Melissodes*, produced by ground-level pan traps was generally low [98]. One possible explanation for this was that larger-bodied bees with strong flying abilities were able to escape from the shallow trapping matrix of the pan traps [99]. Our study supported this observation, as blue vane traps, which were effective at capturing larger bees and exhibited higher functional diversity compared to other trap types. The incorporation of blue vane traps helped to compensate for the shortfall in capturing specific species of pan traps, thereby enriching the functional diversity of the bee community captured in the forests. Another possibility might be the complexity of forest structures that the abundant flowers there might lead to potential competition with traps. For instance, the efficiency of pan traps decreased with the increased floral resources due to the competition between florals and traps [21,30].

### 4.4. Taxonomic, Phylogenetic, and Functional Diversity

The comprehensive capture of bee communities is crucial for the foundation of effective management and conservation programs. In this study, we evaluated not only taxonomic, but also phylogenetic and functional diversity among four trap types to compare the effectiveness of each trap type. It has been shown that sampling methods are biased in terms of functional traits for some arthropods, which was also detected here [27,31]. However, it is important to recognize that the functional traits we focused on might only reveal select ecological functions, potentially overlooking other traits with other ecological functions. For this reason, it is important to incorporate phylogeny into analyses, as phylogenetic diversity has long been invaluable in deciphering ecological forces acting on communities, due to its capability to capture aspects of niche use [43,44,45].

The comparisons among different trap types differed across three dimensions of diversity indices, which emphasizes the importance of involving various dimensions of diversity to accurately assess the effectiveness of various trap types. Interestingly, our results showed that greater taxonomic and phylogenetic diversity was not always correlated with higher functional diversity. For instance, while blue pan traps showed the highest levels of taxonomic and phylogenetic diversity, it was the blue vane traps that yielded the highest functional diversity. This could be explained by the dominant genus captured in blue pan traps, *Lasioglossum*, which tended to be morphologically similar and explore similar resources with each other taxon belonging to this genus. These bees observed in our study were characterized by small body size, nesting below ground, and solitary behavior, consistent with our findings that pan traps exhibited a preference for specific functional groups. Despite the high taxonomic diversity, the prevalence of such similar traits resulted in a lower level of functional diversity.

## 5. Conclusions

Overall, our results highlighted the importance of conducting surveys with diverse trap types to characterize bee fauna. For our sub-tropical forest type, we found that blue vane and pan traps were an efficient combination for bee sampling. Unexpectedly, the combination of three trap types—blue vane, blue pan, and yellow pan traps—yielded a more comprehensive fauna than any of the two-method or three-method combinations. Our findings also emphasized that it was important to include multiple dimensions of diversity, such as taxonomic, phylogenetic, and functional aspects. The comprehensive diversity indices provided a more nuanced picture of community structure and a deeper understanding of composition shifts. The bias to different groups was mostly amplified in mixed forests and the efficiency of trap types varied with tree diversity, indicating the importance of considering habitat types when selecting trapping strategies for bee diversity surveys. Unfortunately, the exact mechanisms of how forest diversity levels affected the sampling bias and effectiveness remained unclear. It also remains unclear whether the efficiency across different traps varied throughout the day, and if so, which potential environmental factors might influence the performance of traps. Furthermore, it would be informative to investigate how seasonal changes in bee activity affect the efficiency of traps. In conclusion, our findings are invaluable for performing an effective and long-term monitoring of bee diversity in the forest. This, in turn, is the basis for a more effective management of bees.

## Figures and Tables

**Figure 1 insects-15-00909-f001:**
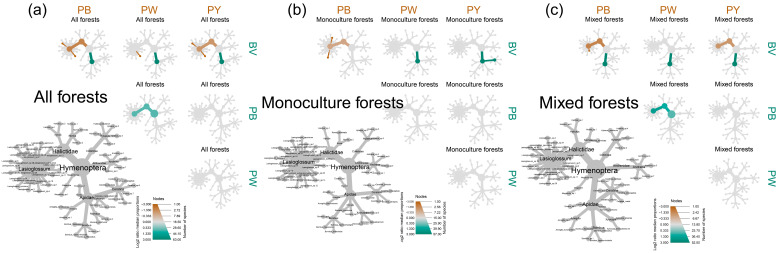
Pairwise comparisons of bee species richness and composition among (**a**) all forests (monoculture and mixed plots), (**b**) monoculture forests, and (**c**) mixed forests. The grey trees of the lower left of each subplot show the complete taxonomy. Smaller trees depict taxonomic differences between trap types. Branches in brown denote higher species richness of those of the column, and green indicate higher species richness across trap types shown on rows. The node colors represent the difference among compared trap types evaluated by log2 ratio of median proportions and the node size represents the number of bee species at each taxonomic level. Abbreviations: PB, blue pan trap; PW, white pan trap; PY, yellow pan trap; and BV, blue vane trap.

**Figure 2 insects-15-00909-f002:**
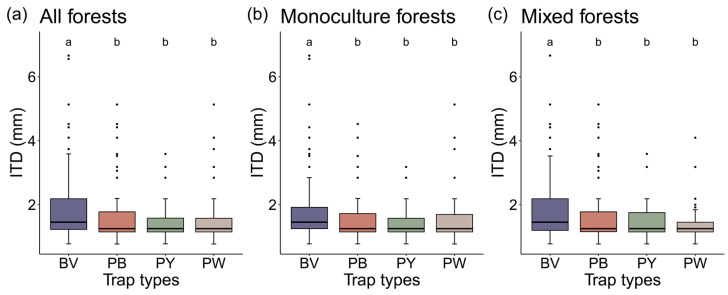
Comparisons of inter-tegular distance (ITD) of bees across four trap types in (**a**) all forests (monoculture and mixed plots), (**b**) monoculture forests, and (**c**) mixed forests. Circles indicate outliers. Letters represent statistical differences according to one-tailed pairwise Wilcoxon test for non-parametric data, with groupings denoted by shared letters (*p* > 0.05). *p* values were adjusted by the Benjamini–Hochberg (BH) method. Abbreviations: PB, blue pan trap; PW, white pan trap; PY, yellow pan trap; and BV, blue vane trap.

**Table 1 insects-15-00909-t001:** Comparisons of three facets of bee diversity in all forests obtained from one-tailed pairwise Wilcoxon test.

Group1	Group2	Counts1	Counts2	Statistic	Counts1	Counts2	Statistic	Counts1	Counts2	Statistic
		Taxonomic			Phylogenetic			Functional		
PB	BV	162	162	**6592.50** ******	162	162	**7480.00** ******	105	92	498.00
PB	PY	162	162	**7756.00** ******	162	162	**9484.00** ******	105	110	**1726.00** ******
PB	PW	162	162	**8752.50** ******	162	162	**9067.00** ******	105	100	**1372.00 ****
BV	PB	162	162	1922.50	162	162	4301.00	92	105	**1155.00** ******
BV	PY	162	162	**4567.50** *****	162	162	**6840.50** ******	92	110	**977.00** ******
BV	PW	162	162	**5606.00** ******	162	162	**6508.00** ******	92	100	**677.00** ******
PY	PB	162	162	890.00	162	162	1542.00	110	105	620.00
PY	BV	162	162	2813.50	162	162	3170.50	110	92	248.00
PY	PW	162	162	**4092.00** ******	162	162	4141.00	110	100	455.00
PW	PB	162	162	427.50	162	162	1664.00	100	105	281.00
PW	BV	162	162	1654.00	162	162	2672.00	100	92	226.00
PW	PY	162	162	1794.00	162	162	3240.00	100	110	406.00

Significant difference is indicated in bold; ** denotes *p* < 0.01; * denotes *p* < 0.05.

## Data Availability

The data presented in this study are available on request from the corresponding author due to we are using some information from this database for other publications.

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
