# Peer review of "Blue Vane and Pan Traps Are More Effective for Profiling Multiple Facets of Bee Diversity in Subtropical Forests"

_insects, 2024, doi:10.3390/insects15110909_

Round 1
Reviewer 1 Report
Comments and Suggestions for Authors
Lines 45-46: I’m not sure what this sentence is trying to say. Could you please rewrite it?
Lines 73-74: This is a great point and I think forests are undersampled as a result.
Line 107: What do you mean by “native sites”?
Lines 115-116: This sentence is a little confusing: what do you mean by “taxonomic, functional groups” separate from the three types of diversity mentioned later in the sentence? Ah, it makes more sense after reading the hypotheses that come after, but I still think this sentence (lines 115-116) should be rewritten to improve its clarity.
Lines 190-193: Can you clarify what is meant here? I read it as saying that you discarded some traps/plots where no bees were caught, but wouldn’t these data points be important in your analysis because bees chose not to visit those traps/plots?
Line 236: What percentage or proportion of the bee community in this region are categorized as aboveground nesters?
Lines 351-352: This is true: passive lethal capture is necessary for any studies where you need a thorough list of the taxa that are present. But, I would appreciate it if the authors could add a note of caution that these passive methods kill large numbers of insects, to the degree that there is some evidence in the literature that passive trapping programs may be negatively affecting populations. A useful citation is https://academic.oup.com/ee/article-abstract/46/3/579/3097253 (Gibbs, J., N. K. Joshi, J. K. Wilson, N. L. Rothwell, K. Powers, M. Haas, L. Gut, D. J. Biddinger, and R. Isaacs. 2017. Does passive sampling accurately reflect the bee (Apoidea: Anthophila) communities pollinating apple and sour cherry orchards? Environ. Entomol. 46: 579–588.)
Reviewer 2 Report
Comments and Suggestions for Authors
insects-3258943 Reviewer comments
Manuscript insects-3258943: .Blue traps outperform for assessing multiple facets of bee diversity in subtropical forests
The manuscript is very interesting. The authors assessed sampling bias of several traps commonly used in pollinator monitoring: blue, yellow, and white pan traps, and blue vane traps, towards different taxonomic and functional groups and their efficiency in measuring taxonomic, phylogenetic and functional diversity. Analyses were performed in monoculture and mixed forests to understand environmental context of trap efficiency. The authors found that blue pan traps generally outperformed other trap types in bee capture and exhibited preference for Halictidae bees. Blue pan traps yielded the highest species richness and phylogenetic diversity, while blue vane traps captured the highest functional richness. Bias differences were frequently detected in mixed forests compared with monoculture forests. The authors found the combination of blue vane and pan traps consistently correlated highest with complete survey among two-method combinations.
The uniqueness of the text is 90% by antiplagiarism.net
The experimental methods and statistics are correct.
The English is almost good but need some correction by native speaker..
The manuscript presents a well-structured study comparing different trap types for bee diversity monitoring, with a strong emphasis on taxonomic, phylogenetic, and functional diversity. It offers new insights into how different traps, especially blue pan and blue vane traps, capture various aspects of bee diversity in subtropical forest ecosystems. The results are promising, with potential applications in conservation strategies and bee community monitoring. However, some areas require improvement
There are some mistakes and comments:
Line 35 - affect - should be - affects.
Line 60 - long term - should be - long-term.
Lines 127-129 - the sentence - In our study, a total of 58 plots from site A and B were involved, including 28 monoculture plots with one tree species in each plot and 30 plots with two or more tree species per plots. - Consider rewording for clarity. maybe like this - Our study included a total of 58 plots from sites A and B, including 28 monoculture plots with one tree species per plot and 30 plots with two or more tree species per plot.
Line 283 - bule pan - should be - blue pan.
Could you perhaps elaborate on the selection criteria used for blue, yellow, and white traps?
Did the species' catches from various traps differ in terms of overlap or redundancy? How did the traps work in concert with one another?
Was there a specific time of day that was shown to affect the efficacy of trapping? If so, what was taken into account?
Have you thought about any potential relationships between the traps themselves, such as how the arrangement of the traps can impact capture?
What made blue traps so much more appealing to Halictidae bees than to other types? Exist any theories or workings behind this preference?
How did the research take into consideration any seasonal differences in bee activity between the times of sampling?
Could you elaborate on the specifics of how the traps were placed inside each plot to prevent biases resulting from plot placement?
The rarefaction analysis demonstrated sufficient sampling completeness. Have you considered using any other sample completeness criteria?
What role did bee sociality and nesting preferences play in the analysis, and will other attributes be examined in later research?
There is currently a restricted selection of functional qualities (sociality, nesting site, and parasitism). Additional features, such as food breadth or proboscis length, might be included in future research to provide more insights.
Please improve the manuscript according to the above comments.
Please check again English style.
